# *Lutzomyia longipalpis* Antimicrobial Peptides: Differential Expression during Development and Potential Involvement in Vector Interaction with Microbiota and *Leishmania*

**DOI:** 10.3390/microorganisms9061271

**Published:** 2021-06-11

**Authors:** Erich Loza Telleria, Bruno Tinoco-Nunes, Tereza Leštinová, Lívia Monteiro de Avellar, Antonio Jorge Tempone, André Nóbrega Pitaluga, Petr Volf, Yara Maria Traub-Csekö

**Affiliations:** 1Laboratório de Biologia Molecular de Parasitas e Vetores, Instituto Oswaldo Cruz, Fiocruz, Av. Brasil 4365, Rio de Janeiro 21040-360, Brazil; erich.telleria@gmail.com (E.L.T.); croookes@gmail.com (B.T.-N.); liviaavellar895@gmail.com (L.M.d.A.); tempone@ioc.fiocruz.br (A.J.T.); pitaluga@ioc.fiocruz.br (A.N.P.); 2Department of Parasitology, Faculty of Science, Charles University, Viničná 7, 12844 Prague, Czech Republic; Terka.Kratochvilova@seznam.cz (T.L.); volf@cesnet.cz (P.V.)

**Keywords:** *Lutzomyia longipalpis*, antimicrobial peptides, innate immunity, RNAi gene silencing, *Leishmania*, microbiota

## Abstract

Antimicrobial peptides (AMPs) are produced to control bacteria, fungi, protozoa, and other infectious agents. Sand fly larvae develop and feed on a microbe-rich substrate, and the hematophagous females are exposed to additional pathogens. We focused on understanding the role of the AMPs attacin (Att), cecropin (Cec), and four defensins (Def1, Def2, Def3, and Def4) in *Lutzomyia longipalpis,* the main vector of visceral leishmaniasis in the Americas. Larvae and adults were collected under different feeding regimens, in addition to females artificially infected by *Leishmania infantum*. AMPs’ gene expression was assessed by qPCR, and gene function of Att and Def2 was investigated by gene silencing. The gene knockdown effect on bacteria and parasite abundance was evaluated by qPCR, and parasite development was verified by light microscopy. We demonstrate that *L. longipalpis* larvae and adults trigger AMPs expression during feeding, which corresponds to an abundant presence of bacteria. Att and Def2 expression were significantly increased in *Leishmania*-infected females, while Att suppression favored bacteria growth. In conclusion, *L. longipalpis* AMPs’ expression is tuned in response to bacteria and parasites but does not seem to interfere with the *Leishmania* cycle.

## 1. Introduction

Antimicrobial peptides (AMPs) are typically cationic peptides with an overall positive charge and hydrophobic amino acid residues [1]. They can kill or neutralize Gram-negative and Gram-positive bacteria, fungi, and parasites [2,3]. There are several possible mechanisms through which AMPs may act, including depolarization of the bacterial membrane [4] and the creation of pores that cause loss of cellular contents [5].

In insects, one classical and well-known aspect of immunity is fat body AMPs synthesis and release into hemolymph. This response in *Drosophila* is coordinated by the Toll and IMD pathways, which are the major immunity regulatory pathways in this insect [6]. Conserved microbial molecular patterns are quickly recognized by pattern recognition receptors which initiate complex intracellular signaling cascades. The innate immune recognition triggers the formation of multi-protein complexes that include kinases and transcription factors, among other regulatory molecules, and culminates in AMP expression [7,8]. Different AMPs can be expressed concomitantly, therefore acting in synergy [9]. Here we focused on three types of AMPs: attacins, cecropins, and defensins.

Attacins form a heterogeneous group of proteins (attacin/sarcotoxin-II family) that vary in size but share common glycine-rich residues. The majority of attacins are active against *Escherichia coli* and other Gram-negative bacteria [10]. Attacins have also antiparasitic activity. The tsetse fly *Glossina morsitans* attacin-A1 acts against *Trypanosoma brucei* bloodstream and procyclic forms in in vitro assays, reducing parasite survival in the fly midgut when added to the infective blood meal [11]. Cecropins are peptides with 31–39 residues with amidated C-termini and a linear α-helical structure without cysteine residues [12]. They display broad-spectrum activity against Gram-negative and Gram-positive bacteria [13,14], and fungi [15]. Synthetic cecropin or cecropin-like peptides cause a reduction of *Plasmodium* ookinetes development in in vitro assays [16,17]. It also reduces oocysts in *Anopheles albimanus* when the peptide is added to infective blood meal [16] or injected in the thorax of previously infected *Anopheles gambiae* [18]. Insect defensins are small cationic peptides of 34–51 residues with six conserved cysteines, and three intramolecular disulfide bonds [19]. Defensins acts on a diverse range of bacteria, usually more efficiently toward Gram-positive than Gram-negative organisms [20]. They also affect fungi and eukaryotic cells. They are detected in the insect hemolymph soon after infection or injury [21] and can act instantaneously to kill bacteria [22]. *Aeschna cyanea* (dragonfly) and *Phormia terranovae* (flesh fly) defensins can profoundly affect *Plasmodium gallinaceum* parasites, reducing oocysts number in *Aedes aegypti* and altering sporozoite morphology with consequent loss of motility [23].

Sand flies (Diptera: Psychodidae, Phlebotominae) are vectors of viruses, bacteria, and protozoans that cause diseases of public health importance [24]. Several species belonging to the *Lutzomyia* and *Phlebotomus* genera are proven vectors of human leishmaniasis and more than 1 billion people are at risk of acquiring the disease [25]. Female sand flies ingest *Leishmania* parasites together with blood when feeding on an infected host. Inside the sand fly gut, *Leishmania* parasites undergo a sequence of multiplication and differentiation steps that culminate with the development of an infective form which can be transmitted to another host through the sand fly bite [26,27]. During this process, the vector immunity is regulated to control microbial challenges [28].

Early reports on sand fly AMPs identified a cecropin-like peptide in *L. longipalpis* [29], and a defensin peptide in *Phlebotomus duboscqi* [30] hemolymph after being injected with bacteria. Later, a defensin gene was identified in *L. longipalpis* [31] and the expression of this gene was shown to be upregulated in females fed with Gram-negative and Gram-positive bacteria [32]. Although earlier studies identified AMPs in adult sand flies, three other AMPs, an attacin (Att), a cecropin (Cec), and a second defensin (Def2), were shown to be modulated in *L. longipalpis* LL5 embryonic cells under microbial challenges, possibly regulated by Toll and IMD pathways concomitantly [33]. Having this set of AMPs identified in *L. longipalpis*, we hypothesized first that these AMPs were expressed either to balance or control the presence of commensal or possibly harmful microbes (in larvae or adult sand flies) and *Leishmania* (in female sand flies); and, second, that the suppression of AMPs by RNAi-mediated gene silencing could influence the microbiota and *Leishmania* development in the female. In the present work, we addressed these hypotheses by selecting different feeding regimens of larvae and adult sand flies and evaluated AMP gene expression levels. We also selected two AMPs (Att and Def2) for silencing in female sand flies followed by *Leishmania* artificial infection, and assessed the parasite and microbiota abundance using qPCR, and parasite development using light microscopy techniques.

## 2. Materials and Methods

### 2.1. AMPs Sequences

*L. longipalpis* attacin (Att) (GenBank KP030755), cecropin (Cec) (GenBank KP030754), defensin 1 (Def1) (GenBank EF491251), and defensin 2 (Def2) (GenBank KP030758) were identified in previous studies [31,33]. Defensin 3 (Def3) (MW269862) and defensin 4 (Def4) (MW269863) were identified using hidden Markov models (HMMER) of insect defensins as a query for a blastp analysis on an unassembled genomic sequence database of *L. longipalpis* (GenBank PRJNA20279). *L. longipalpis* defensin amino acid sequences were analyzed using the InterPro Classification of Protein Families 81.0 tool [34]. Similarities of *L. longipalpis* AMP amino acid sequences with other insects were assessed by ClustalW multiple alignment tool [35], followed by phylogenetic analysis using MEGA6 software [36], with the maximum likelihood method and the Le_Gascuel_2008 model [37]. Gamma distribution was used to model evolutionary rate differences among sites and bootstrapping with 100 replicates.

### 2.2. Lutzomyia Longipalpis

*L. longipalpis* larvae and adult stages were obtained from a previously established colony originally collected in Jacobina, BA, Brazil, and kept under standard insectary conditions at temperatures between 24 and 28 °C and 70–80% relative humidity [38]. For colony maintenance, adult insects were fed on 50–70% sucrose *ad libitum*, and females were blood-fed on anesthetized hamsters or mice once per week. For experimental procedures, sand flies were collected as follows. Larvae at 3rd (L3), 4th (L4), or pre-pupae instars were collected from rearing pots and cleaned with thin dry brushes. The first and second larval stages were not collected due to their diminutive size. Emerged male and female sand flies (1 to 3 days old) were released into rearing cages and non-fed insects were collected immediately prior to offering sugar-meal. Sugar-fed insects were collected at 24 and 48 h post-feeding (PF) ad libitum on sugar-meal containing blue aniline to positively identify fed insects. Female sand flies (3 to 6 days old) were artificially fed through chick skin membrane on rabbit blood seeded with *Leishmania infantum* (10^6^ parasites/mL of blood), while control groups were fed on blood without parasites under the same conditions. Fully engorged females were separated and collected at different times PF (24, 48, 72, and 96 h) for RNA extraction and microscopy analysis (192 h). All samples were collected in pools of 10 larvae or adult sand flies.

### 2.3. RNA Extraction and cDNA Synthesis

Larvae or adult sand flies pools were collected at different stages or timepoints PF (described above) for total RNA extraction using TRIzol™ Reagent (Invitrogen, Carlsbad, CA, USA) according to the manufacturer’s instructions. Extracted RNA was incubated with RNase-free DNase I (Thermo Scientific, Carlsbad, CA, USA) at 1 U/μg of total RNA for removing possible traces of genomic DNA. Up to 1 μg of total RNA was used in reverse transcriptase reactions to produce cDNA using SuperScript III Reverse Transcriptase (Invitrogen). Protocols were followed according to the manufacturers’ instructions.

### 2.4. RNAi-Mediated Gene Silencing

Gene-specific primers (dsAtt-F, dsAtt-R, dsDef2-F, and dsDef2-R) coupled to a T7 promoter sequence (Appendix A) were designed to amplify templates from sand fly cDNA by PCR. dsLacZ-F and dsLacZ-R primers were used to amplify the template from p-GEM-T Easy plasmid (Promega) as control dsRNA. Touchdown PCR was used as follows: 95 °C for 3 min; 16 cycles of 95 °C for 45 s, 68 to 50 °C (progressively decreasing 1 °C per cycle) for 45 s, and 72 °C for 45 s; 26 cycles of 95 °C for 45 s, 50 °C for 45 s, and 72 °C for 45 s; 72 °C for 3 min. These templates were purified by Wizard SV Gel and PCR cleanup system (Promega) and used in dsRNA synthesis reaction by MEGAscript RNAi kit (Invitrogen) following the manufacturer’s instructions. The produced dsRNA was lyophilized and resuspended in ultrapure H_2_O to 4.5 μg/μL final concentration. Sand flies were microinjected intrathoracically with 32.2 nL of dsRNA using Nanoject II microinjector (Drummond) [39].

### 2.5. Gene Expression Analysis by qPCR

The gene expression was assessed by qPCR using cDNA templates, gene specific-primers (Appendix A), and SYBR Green PCR Master Mix in a 7500 Real-Time PCR System (Applied Biosystems) following the manufacturer’s standard cycling conditions. The gene expression was calculated relative to a ribosomal protein (RP49) reference gene and expressed in fold change values in comparison to a control group [33] following the ΔΔC_T_ method [40].

### 2.6. Leishmania Development in Sand Fly Guts

On day 8 post-infection, a minimum of 20 sand fly guts were examined by light microscopy for parasite load, localization, and development. Guts were dissected in saline solution (NaCl 0.9%) and examined under a 40× magnification objective lens. Parasite loads were estimated and classified as low (below 100 parasites), medium (between 100 and 1000 parasites), or heavy infection (above 1000 parasites) [41]. The parasite localization throughout the gut (abdominal or thoracic gut, cardia, and colonized stomodeal valve) was recorded [41,42]. In addition, parasite developmental stages were inspected on sand fly gut smears on Giemsa-stained glass slides under a 100× magnification objective lens. Images of 100 randomly selected promastigotes were captured, and cell width, length, and flagellum were measured using the microscope scale plugin in ImageJ 1.52a software [43]. Parasites were categorized as elongated nectomonads (body length ≥ 14 μm), procyclic promastigotes (body length < 14 μm and flagellar length ≤ body length), metacyclic promastigotes (body length < 14 μm and flagellar length ≥ 2× body length), leptomonads (remaining parasites) [42,44]. Samples were collected from a minimum of 3 independent experiments.

### 2.7. Statistical Analysis

The Kruskal–Wallis test with Dunn’s correction for multiple comparison was used to calculate significant differences in gene expression levels of larval stages L3 and L4 in comparison to pre-pupal stages, and differences of adult sand flies (males and females) fed on sucrose collected at 24 and 48 h in comparison to non-fed sand flies.

Ordinary two-way ANOVA with Sidak’s correction for multiple comparisons test was used to calculate significant differences in: (a) gene expression levels in *Leishmania*-infected in comparison to blood fed females, both collected at several time points post-feeding; (b) gene expression levels between Att and Def2 dsRNA injected in comparison to LacZ dsRNA injected females collected at several time points post-injection, and (c) infection estimation and localization and parasite morphology in samples collected from Att and Def2 dsRNA injected in comparison to LacZ dsRNA injected females. The statistical analysis was carried in GraphPad Prism software (version 6.07) (GraphPad Software Inc., San Diego, CA, USA).

## 3. Results

### 3.1. AMP Sequences

Phylogenetic analysis showed that *L. longipalpis* Att amino acid sequence [33] is closely related to the sequences of the sand flies *Nyssomyia neivai* (GenBank JAV08575) and *P. papatasi* (VectorBase PPAI003791-RA), forming a separate group from other insect sequences (Appendix A). Cec [33] forms a group with *P. papatasi* (VectorBase PPAI003330-RA), separated from the *A. gambiae* sequence (Appendix A).

Def1 and Def2 were previously identified [31,33] and in the present work we described two other defensin sequences Def3 and Def4 that contain the six conserved cysteine residues (Appendix A) characterizing these AMPs (InterPro IPR001542) [34]. The four *L. longipalpis* defensins contain an arthropod defensin-2 superfamily motif (pfam01097) [34]. Def3 and Def4 share 70.45% and 53.93% of similarity with Def1, respectively, and they both share less than 50% similarity with Def2 (Appendix A). Phylogenetic analysis showed that Def1 and Def3 grouped within a same branch. Def4 forms a sister branch that includes a *P. papatasi* sequence (VectorBase PPAI004255-RA). Def2 forms a group with the *N. neivai* defensin 2 sequence (GenBank JAV13023) and is more distantly related to the other *L. longipalpis* defensins (Appendix A).

### 3.2. AMP Expression in Larval Stages

Sand fly larvae develop and feed on a microbe-rich substrate which could impact AMP expression. We evaluated the expression levels of attacin, cecropin and four defensin genes in *L. longipalpis* L3 and L4 instar larvae, in comparison to the pre-pupal (PP) stage that interrupts feeding prior to pupation and defecates the midgut content. There is a significant increase in Att, Def2, and Def4 in L3 stages (Figure 1A,D,F), whereas Cec, Def1, and Def3 do not show a significant difference among these stages (Figure 1B,C,E). Bacteria loads estimated by the 16S ribosomal RNA primers showed that L3 and L4 stages harbor more bacteria than the PP stage (Figure 1G).

### 3.3. AMP Expression in Sugar-Fed Adult Sand Flies

After the emergence of imagoes, insects were fed a sucrose rich solution, which can stimulate gut microbiota growth. Upon sugar intake, these insects might also modulate their AMP expression in response to the microbial community growth. We observed that, in males, Att, Cec, Def2, and Def4 are significantly increased at 48h PF (Figure 2A,B,D,F). Def1 and Def3 are highly expressed at 24 and 48 h in comparison to non-fed males (Figure 2C,E). In females, Att and Cec are reduced at 48 h (Figure 2G,H), whereas Def3 is reduced at 24 and 48 h PF (Figure 2K). Def1 increased at 24 and 48 h (Figure 2I), whereas Def2 showed no significant difference when compared to non-fed females (Figure 2J). Bacteria loads in males were reduced at 48 h (Figure 2M) and increased in females after feeding on sucrose at 24 and 48 h PF (Figure 2N).

### 3.4. AMP Expression in Leishmania-Infected Sand Flies

Addressing the potential modulation of *L. longipalpis* AMPs in the presence of *Leishmania*, we investigated the AMP expression of female *L. longipalpis* artificially fed a parasite-seeded blood-meal in comparison to a control group fed blood. Att and Def2 expressions were significantly increased at 72 h (Figure 3A,D) and Cec at 48 and 72 h (Figure 3B), whereas Def1, Def3, and Def4 showed no significant changes (Figure 3C,E,F). We also evaluated the *Leishmania* and bacteria content in the infected insects. We observed a parasite load reduction at 72 h and an increase in bacteria at 48 h (Figure 3G,H).

### 3.5. AMP Gene Silencing Followed by Leishmania Infection

Considering the Att and Def2 significant expression increases in *Leishmania* infected females, we hypothesized that RNAi-mediated gene silencing of these genes could interfere with the parasite growth in artificially infected sand flies. We first tested if Att- and Def2-dsRNAs would silence their corresponding genes using recently emerged flies kept under colony conditions fed sucrose solution. Both genes were successfully silenced at 24 and 48 h post-dsRNA injection when compared to the control group injected with LacZ dsRNA (Appendix A). For testing Att and Def2 gene silencing effect in *Leishmania* infection, insects were injected with dsRNA and artificially fed an infective blood meal on the following day. Att-dsRNA significantly reduced Att gene expression from 24 to 72 h post-infection (Appendix A), and Def2-dsRNA reduced Def2 gene expression from 48 to 72 h post-infection (Appendix A).

The parasite load in Att-silenced group remained similar to the control group from 24 to 72 h PF (Figure 4A). We also investigated the effect of Att silencing in the insect microbiota. There was a significant increase in the bacteria levels at 72 h post-infection (Figure 4B). In Def2-silenced sand flies, we observed a slight but non-significant variation of *Leishmania* detection on first- and third-days post-infection (Figure 4C) and non-significant bacterial increase on the first day post-infection (Figure 4D).

### 3.6. Leishmania Late Infection in RNAi-Silenced Sand Flies

To explore Att and Def2 silencing effect on the parasite development and colonization progress in the insect gut, we estimated the infection load, assessed the parasite localization and morphology in *L. longipalpis* gut at 192 h post-infection using light microscopy. No significant differences were detected in infection estimation levels in Att- or Def2-dsRNA-injected sand flies in comparison to the LacZ-dsRNA-injected control group (Figure 5A). There was a non-significant increase in thoracic gut localization in Def2-dsRNA-injected groups (Figure 5B). We also evaluated the parasite morphology in Giemsa-stained gut smears, and no significant difference was observed between the dsRNA-injected groups (Figure 5C).

## 4. Discussion

We have previously identified and characterized an attacin (Att), a cecropin (Cec), and two defensin genes (Def1 and Def2) [31,32,33] in *L. longipalpis*. In the present study, we investigated the expression of these AMP genes, in addition to two other defensin genes (Def3 and Def4). 

Phylogenetic studies show that different sand flies’ Att sequences are closely related to each other and are less similar to other vectors, such as *G. morsitans* and *Aedes albopictus*. Similarly, phlebotomine Cec sequences are closely related to each other and grouped separately from other insects. The close similarities of Att or Cec between sand fly species can be explained by the natural evolution resulting from a common ancestor. Nevertheless, Def3 is closely related to Def1, whereas Def2 shares fewer similarities with the other *L. longipalpis* defensins. An additional evolutionary pressure appears to occur in these genes. For instance, defensins from different organisms were classified into two superfamilies as a result of independent evolutionary origins, with structural and functional similarities. A larger group consists of cis-defensins found in plants, fungi, and invertebrates, which includes the *L. longipalpis* defensins. The smaller group consists of trans-defensins from vertebrate and invertebrate defensins that contain eight conserved cysteines. Each of these superfamilies went through subsequent divergent evolution [45].

In nature, *L. longipalpis* larvae develop in soil sites that generally contain organic matter derived from decaying plants and animal droppings [46,47]. These environments often harbor a diversity of microorganisms [48,49,50] that can be ingested when larvae feed on the available substrates, further contributing to the diversity of the sand fly gut microbiome [28]. In *P. duboscqi* and *Lutzomyia evansi*, microbial community varies depending on developmental stage [51,52,53], indicating that breeding sites and feeding habits together influence microbiota diversity and abundance.

Our present results regarding *L. longipalpis* larvae AMP gene expression showed Att, Def2, and Def4 up modulation in L3 but not in L4 larval stages in comparison to the non-feeding pre-pupal (PP) stage. This finding may be related to highly active and voracious feeding habits of sand fly L3 stage [54,55]. Nevertheless, bacterial loads in both L3 and L4 stages are higher, suggesting bacterial abundance is not the only factor influencing AMPs regulation. It is possible that gut bacterial diversity changed through these developmental stages and the insect adjusted expression of a specific set of AMPs to control the gut microbial community. It is interesting to note that *Bacillus subtilis* or *Pantoea agglomerans* artificial feeding caused a reduction in Att expression in *L. longipalpis* L3 larvae at 12 h PF, whereas Def1 was increased at 24 h after feeding on *P. agglomerans* [54]. These findings indicate that the larvae immune system may be tuned according to a diverse microbiome and distinguish bacterial strains to support larval development.

The gut microbial load control allows a reduced set of bacteria, such as *Ochrobactrum* sp., to be carried from larval to pupae and adult sand flies [51,56,57]. This finding may reflect the adaptative advantage in hosting commensal or symbiotic bacteria that will protect the insect host from pathogenic infections while provide essential nutrients [58]. Therefore, it is plausible to consider that newly emerged *L. longipalpis* successfully balance the microbial exposure throughout metamorphosis. As probing surfaces resume, sand flies are exposed to new sources of microbes [51]. Moreover, the newly ingested nutrients may be beneficial to the remaining or resident bacteria. For instance, both male and female sand flies feed on plant-derived food sources such as nectar, honeydew, or phloem sap in nature [59,60,61,62], but under laboratory conditions, a sucrose-rich solution is used as a substituent of the carbohydrate-rich source from plants. Therefore, during sucrose feeding, a small amount of the ingested sucrose first passes through the stomodeal valve, and then a larger amount is stored in the crop [63]. As a result, the carbohydrate nutrient is readily available in the gut and to the resident bacteria.

We investigated the AMP expression in adult sand flies (males and females) after emergence (non-fed) and after sucrose feeding. In *L. longipalpis* males, AMP expression was increased either on the first- or second-day post-sucrose feeding which reflects the need to balance the microbial community in the gut. Because they feed exclusively on sucrose-rich food, they may have adapted to a stronger antibacterial immune response to control the growth of carbohydrate-related bacteria in their gut. Alternatively, males could be expressing higher levels of AMPs to transfer them to females during mating, as seen in *Drosophila* males that express andropin [64] in accessory glands or ejaculatory ducts, and transfer it within the seminal fluid [65]. For example, drosomycin is detected in *Drosophila* ejaculatory structures and is likely to be present in the ejaculate [66]. In addition, in *L. longipalpis* sugar-fed females, our results show that the initial response occurs through Def1 expression possibly due to the consequent bacterial growth. Indeed, *L. longipalpis* females fed a sucrose diet showed higher bacterial diversity when compared to blood-fed sand flies [67]. The reduction of Att, Cec, and Def3 reveals a balanced AMP expression and it is plausible to assume that they are adjusted according to the changes in bacteria diversity.

In previous work, we showed that Def1 expression in sand fly females fed different bacteria added to the sucrose meal varied significantly depending on which bacteria were ingested when compared to the control group fed on sucrose only. Def1 was upregulated by *E. coli*, *Ochrobactrum* sp., *Serratia marcescens*, and *M. luteus*, and downregulated by *Pantoea agglomerans* feeding challenges [32], again indicating that the sand fly immune response distinguishes among different pathogens.

No previous data has been published on Att, Cec, and Def2 expression in *L. longipalpis* female under different feeding conditions or under different bacterial challenges. Nevertheless, in LL5 embryonic cells, Att expression increased after *Staphylococcus aureus*, *Serratia marcescens*, and the yeast *Saccharomyces cerevisiae* exposure; Cec increased after *E. coli*; and Def2 increased after *E. coli* and *S. aureus* ingestion [33]. This adjusted sand fly immune response against a variety of bacteria species may be connected to transcription regulators, as seen in *Drosophila*, where regulators can increase or decrease AMPs’ expression [68] with consequent impact on commensal and pathogenic bacteria [69].

*L. longipalpis* is the natural vector of *L. infantum* in the Americas [70], and we were interested to investigate the sand fly AMP expression during natural parasite–vector interaction. Our results show that Att, Cec, and Def2 are all upregulated on the third day after *L. infantum* infection when compared to the blood-fed control group. This time point corresponds to blood meal digestion end when the insect peritrophic matrix is degraded [71], and parasites begin to colonize the gut [72]. Our results show that at this same time point, loads of *Leishmania* are reduced. This finding can be directly related to the increase of AMP levels and/or defecation of blood meal remains [26]. In the present study, we did not address the proportions of each parasite form on the third day post-infection, but previous studies show that at this timepoint nectomonad and leptomonad forms are the most abundant forms [72,73]. Furthermore, the presence of *Leishmania* infection is associated with a reduction in bacterial diversity following the infection progression [28,67,74,75,76]. As infection continues, for example, on the sixth day when nectomonad forms are still the most abundant forms and metacyclic forms are present [72], we previously showed that Def1 was downregulated in *L. longipalpis* infected by *L. mexicana* [32]. The only study that tested a sand fly AMP activity against a set of microbes showed that a *P. duboscqi* recombinant defensin is active against *L. major* promastigotes and some fungi [30]. Therefore, the sand fly immune response in late-stage *Leishmania* development may differ according to the reduced diversity of gut microbiota [67,75].

Although it is not yet known if sand fly immune response is driven specifically toward the parasite or the microbial community, we hypothesized that the suppression of these AMPs could benefit the growth of the parasite, bacterial community, or both. To address this point, we suppressed the function of AMPs by RNAi-mediated gene silencing, for which we selected the two significantly upregulated AMPs during *L. infantum* infection, Att and Def2.

We investigated the parasite load in AMP-silenced insects during three consecutive days after infection by qPCR to assess parasite and bacterial abundance in comparison to the LacZ-dsRNA-injected control group. *L. longipalpis* Att gene silencing resulted in a slight decrease of *L. infantum* and a significant increase of bacteria loads on the third day post-infection. These results indicate that the suppression of Att has a more evident effect on bacteria, and the reduction of the parasite may happen as a consequence of the gut microbiome increase. *Leishmania* detection seems to increase after Def2 is silenced, but the difference is not statistically significant.

We also hypothesized that AMP expression changes, observed in initial days post-gene silencing, could impact parasite development in late phase infection. To address this aspect, we investigated the parasite loads and localization in dissected guts on the eighth day post-infection, a time when *Leishmania* can be already detected in the stomodeal valve [72,73]. Dissected guts were individually inspected to estimate the parasite load and gut localization. Samples from Att and Def2-dsRNA injected flies were compared to the LacZ-dsRNA injected group. Our results, showed no significant differences in the parasite loads.

Although overall parasite loads could represent the ability of the parasites to survive and multiply, the localization of the parasites could yield more detailed information on the ability of the parasite to migrate and complete the cycle in the vector. The dsRNA-injected groups comparison showed no significant differences in parasite localization, but the majority of Def2-silenced insects showed parasites in the thoracic part of the insect gut. It is possible to presume that Def2 suppression resulted in conditions that favored parasites to colonize the anterior segment of the sand fly gut.

It appears that Att silencing creates a less favorable environment in the early phase of infection, possibly due to competition with bacteria, but in the late phase of infection, the parasites were able to colonize the gut similarly to the case of the control group. In addition, Def2 silencing apparently creates a more favorable environment in the early phase of infection, affecting the late phase when parasites are more present in gut anterior segments. These signs of parasite localization in the sand fly gut are tightly associated with the *Leishmania* cycle. Therefore, parasite morphological changes could yield additional information on the infection progress. In our experimental settings, the proportion of metacyclic promastigotes did not alter after Att or Def2 silencing.

AMPs are commonly expressed in synergy to achieve efficient microbe control, and the suppression of Att or Def2 may cause minor changes in abundance or localization of *L. infantum* in *L. longipalpis* gut, but parasites adapted to the changes in the sand fly immune response. How *Leishmania* parasites evade the sand fly immune response remains an intriguing aspect that deserves further investigation.

## Figures and Tables

**Figure 1 microorganisms-09-01271-f001:**
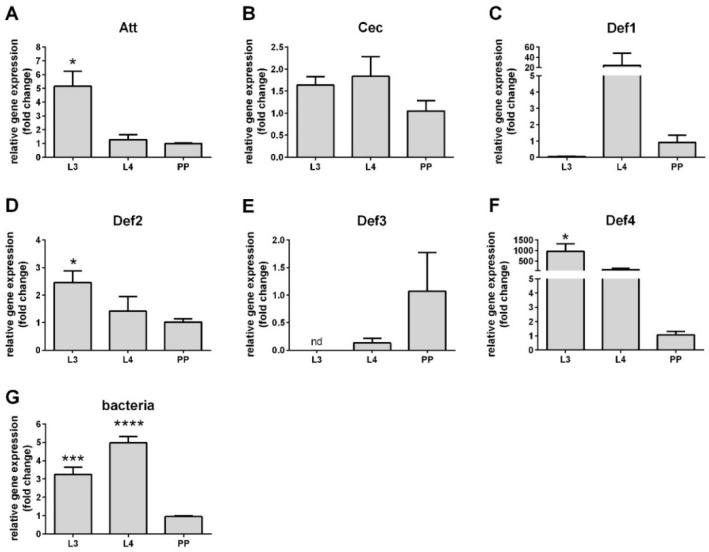
Relative gene expression of antimicrobial peptides (AMPs) in larval stages. (**A**) attacin; (**B**) cecropin, (**C**) defensin 1; (**D**) defensin 2; (**E**) defensin 3; (**F**) defensin 4; (**G**) bacteria 16S. The *y*-axis represents relative expression values expressed in foldchange in comparison to the pre-pupae stage. The *x*-axis represents larval stages L3, L4, and pre-pupal (PP). Vertical bars represent the average values of three independent experiments, and error bars represent the standard error. Asterisks indicate significant differences (* *p* < 0.05; *** *p* < 0.001; **** *p* < 0.0001). (nd) not detected.

**Figure 2 microorganisms-09-01271-f002:**
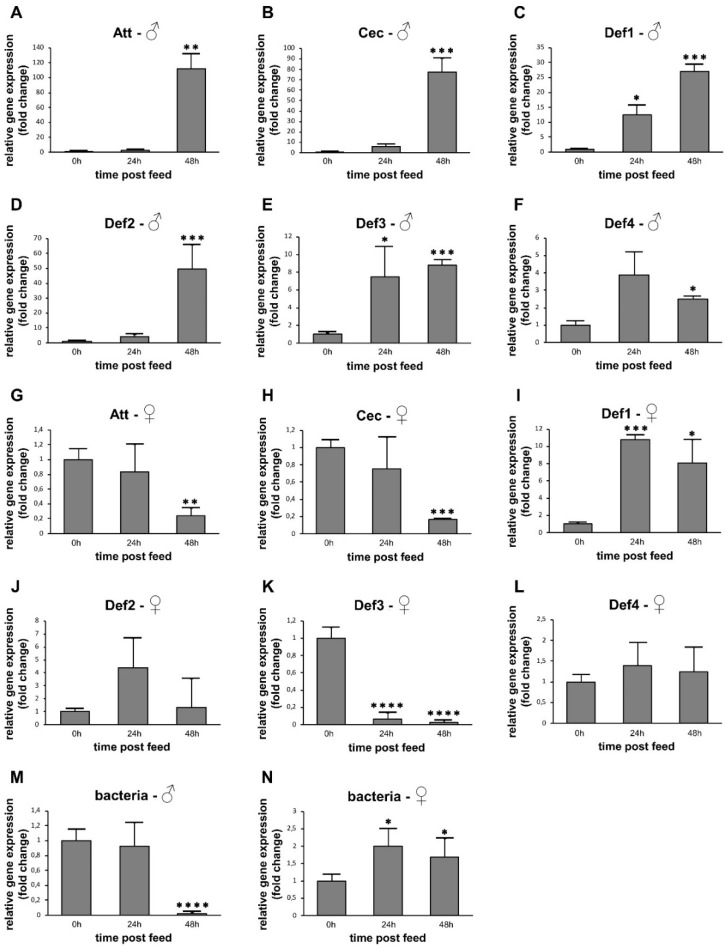
Relative gene expression of AMPs in sugar-fed sand flies. (**A**–**F**,**M**) males; (**G**–**L**,**N**) females; (**A**,**G**) attacin; (**B**,**H**) cecropin; (**C**,**I**) defensin 1; (**D**,**J**) defensin 2; (**E**,**K**) defensin 3; (**F**,**L**) defensin 4; (**M**,**N**) bacteria 16S. The *y*-axis represents relative expression values expressed in comparison to non-fed males or non-fed females. The *x*-axis represents the feeding stage: non-fed (0 h) or sugar-fed collected at 24 and 48 h post-feeding (PF). Vertical bars represent the average values of three independent experiments, and error bars represent the standard error. Asterisks indicate significant differences (* *p* < 0.05; ** *p* < 0.01; *** *p* < 0.001; **** *p* < 0.0001).

**Figure 3 microorganisms-09-01271-f003:**
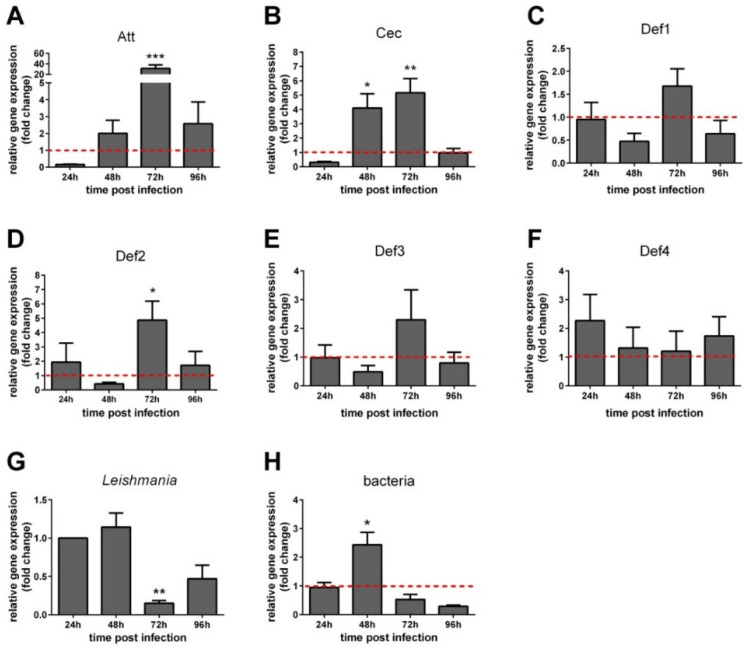
Relative gene expression of AMPs in blood-fed females infected by *Leishmania*. (**A**) Att; (**B**) Cec; (**C**) Def1; (**D**) Def2; (**E**) Def3; (**F**) Def4; (**G**) *Leishmania* actin; (**H**) bacteria 16S ribosomal RNA. The *y*-axis represents relative gene expression as fold change values, in comparison to the control group of blood-fed females (dotted line) collected at the corresponding time points (**A**–**F**,**H**); *Leishmania* detection was expressed in comparison to 24 h (**G**). The *x*-axis represents females collected at first to fourth day PF. Vertical bars represent the average values of three independent experiments, and error bars represent the standard error. Asterisks indicate significant differences (* *p* < 0.05; ** *p* < 0.01; *** *p* < 0.001).

**Figure 4 microorganisms-09-01271-f004:**
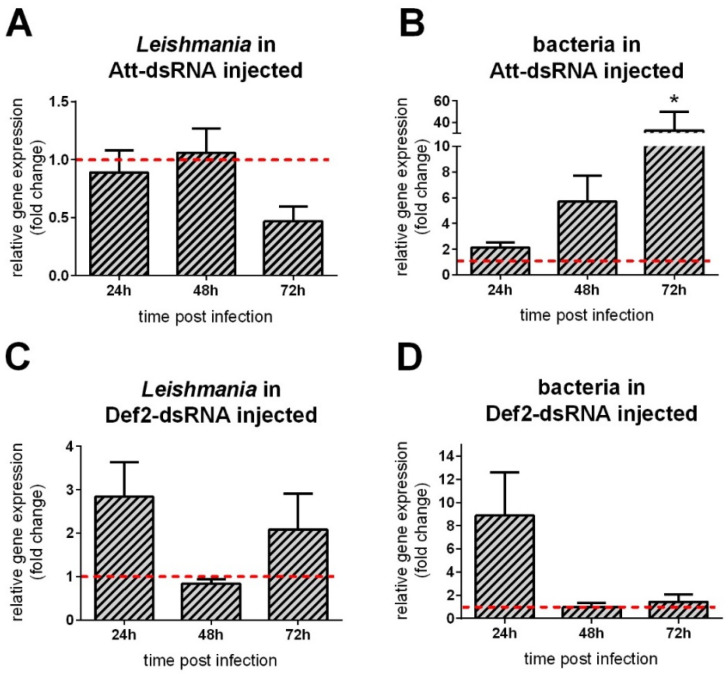
Relative gene expression of AMPs in dsRNA injected females Figure 2. dsRNA-injected; (**A**,**C**) *Leishmania* actin; (**B**,**D**) bacteria 16S ribosomal RNA. The *y*-axis represents relative gene expression as fold change values of Att or Def2-dsRNA injected insects in comparison to the LacZ-dsRNA injected control group. The *x*-axis represents females collected at first to third days post-infectious feeding. Vertical bars represent the average values of three independent experiments, and error bars represent the standard error. Asterisks indicate significant differences (* *p* < 0.05).

**Figure 5 microorganisms-09-01271-f005:**
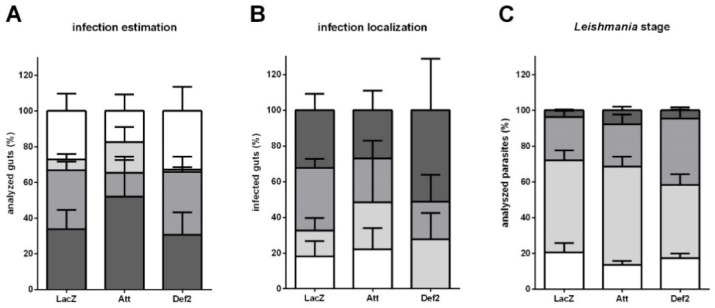
Infection intensity and development at late infection in dsRNA-injected sand flies. (**A**) The *y*-axis represents the percentage of all individually inspected insects (minimum of 20 sand flies in each dsRNA injected group). Bar colors indicate infection intensity: non-infected (white); with low (light grey); medium (mid grey); and heavy (dark grey) infections. (**B**) The *y*-axis represents the percentage in infected insects to measure the infection progress in the gut. Bar colors indicate sand fly gut localization: parasites reached the stomodeal valve (dark grey); cardia (mid grey); thoracic gut (light grey); or stayed in abdominal gut (white). (**C**) The *y*-axis represents the percentage of analyzed parasites to measure parasite development in the sand fly gut. Bar colors indicate parasite developmental stage: metacyclic promastigote (dark grey); leptomonad (mid grey); elongated nectomonad (light grey); procyclic promastigote (white). The *x*-axis represents dsRNA injected groups. Vertical bars represent the average values of three independent experiments, and error bars represent the standard error. No significant differences were found.

## Data Availability

The newly identified *L. longipalpis* defensin genes can be found in GenBank, MW269862 and MW269863.

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
