# Peer review of "Lutzomyia longipalpis Antimicrobial Peptides: Differential Expression during Development and Potential Involvement in Vector Interaction with Microbiota and Leishmania"

_microorganisms, 2021, doi:10.3390/microorganisms9061271_

Round 1

Reviewer 1 Report

The researchers in this study conduct several experiments in which they evaluate the expression of a set of antimicrobial peptides in response to diet and Leishmania presence. My main critique of this study, which influenced my understanding of the entire paper was the lack of rigorous statistical analysis. The authors did not include a description of their statistical analysis in their methods and did not use rigorous statistical analysis of their data- or if they did it was not clear to me. Given the number of genes you were investigating over multiple time periods a linear mixed model that adjusted for multiple comparisons is appropriate. Without this analysis, it is difficult to identify random variation from statistical significance. I think some of your significant comparisons will no longer be significant once your data is properly analyzed and that this will change how you interpret your data. I also think that on several occasions the data does not support your conclusions.

For example

In your results, you state: “that feeding larvae are exposed to a higher bacterial content and express AMPs to control the gut microbial community.”

As written, I do not believe that your data supports this claim. Only 3 of the 6 AMPs (50%) have significantly higher expression in L3 compared to PP. Additionally, although the level of bacteria goes up from L3 to L4 the expression of ATT, Def2, and Def4 goes down in L4 and is not significant from the pp group. This does not support your statement. You do not attempt to explain why expression would go down from L3 to L4 if gene expression is positively influenced by bacterial abundance. 

This critique influences how I view the whole paper, it seems like you are reading into some data points and ignoring others. A rigorous data analysis would alleviate this concern. 

I have other issues such as the use of different scales on the y-axis in your figures- it is really confusing. Consider using a standardized scale for gene expression, but this is a minor edit. 

Author Response

Reply to Reviewer #1

The researchers in this study conduct several experiments in which they evaluate the expression of a set of antimicrobial peptides in response to diet and Leishmania presence. My main critique of this study, which influenced my understanding of the entire paper was the lack of rigorous statistical analysis. The authors did not include a description of their statistical analysis in their methods and did not use rigorous statistical analysis of their data- or if they did it was not clear to me.

For the results shown in figure 1 and figure 2, analyses were made between a test sample versus a control sample using Student t-test tool in GraphPad Prism (version 6.07) which applies Kruskal-Wallis test for non-parametric distribution with Dunn’s correction for multiple comparison.

For results shown in figures 3, 4, 5, and S5, analyses were made between a test group versus a control group (both having multiple timepoints or categories) using Two-Way ANOVA tool in GraphPad Prism (version 6.07) which applies ordinary two-way ANOVA with Sidak’s correction for multiple comparisons test.

We had a short description of statistical analysis described on each figure legend, but to address the reviewer’s comment, we added a paragraph of statistical analysis methods on 2.7.

Given the number of genes you were investigating over multiple time periods a linear mixed model that adjusted for multiple comparisons is appropriate. Without this analysis, it is difficult to identify random variation from statistical significance. I think some of your significant comparisons will no longer be significant once your data is properly analyzed and that this will change how you interpret your data. I also think that on several occasions the data does not support your conclusions.

The mixed effects model approach is a powerful method and can be used to analyze a wide variety of experimental designs, but in the newest version of Graph Pad Prism (version 9.0.0) this option is applied to analyze repeated measures data with missing values. In the case of complete data, ANOVA is still the choice method for analyzing repeated measures. Therefore, we choose to keep our choice of methods for statistical analysis.

For example

In your results, you state: “that feeding larvae are exposed to a higher bacterial content and express AMPs to control the gut microbial community.” As written, I do not believe that your data supports this claim. Only 3 of the 6 AMPs (50%) have significantly higher expression in L3 compared to PP.

Additionally, although the level of bacteria goes up from L3 to L4 the expression of ATT, Def2, and Def4 goes down in L4 and is not significant from the pp group. This does not support your statement. You do not attempt to explain why expression would go down from L3 to L4 if gene expression is positively influenced by bacterial abundance.

We thank reviewer 1 for raising these important points.

We actually did not expect all the AMPs here studied to be modulated homogeneously under a same stimulus. It is well known that different AMPs are expressed upon different challenges in insects. For instance, in a publication that examined the immune responses of L. longipalpis LL5 cells to various challenges, we have shown that different AMPs are modulated by the exposure to, for instance, Leishmania or 4 different bacteria (Tinoco-Nunes B et al. The sandfly Lutzomyia longipalpis LL5 embryonic cell line has active Toll and Imd pathways and shows immune responses to bacteria, yeast and Leishmania. Parasit Vectors. 2016 Apr 20;9:222. doi: 10.1186/s13071-016-1507-4). One of our interests in developing the present work was exactly trying to understand if specific AMP were expressed in infections with Leishmania or with microbiota in general.

To improve clarity, we rephrased the discussion section referring to AMPs expression in larvae. The modified part now reads:

“Our present results on the L. longipalpis larvae AMPs gene expression showed that Att, Def2, and Def4 were increased in L3 but not in L4 larval stages in comparison to the non-feeding pre-pupa (PP) stage. This may be explained by the fact that the sand fly L3 stage is very active and voracious [54,55]. Nevertheless, bacterial detection results showed higher loads in both L3 and L4 stages, which suggests that bacterial abundance is not the only factor influencing these AMPs regulation. It is possible that gut bacterial diversity change through these developmental stages and the insect adjusted expression of a specific set of AMPs to control the gut microbial community.”

It will be of great interest in follow up projects to identify possible changes in microbiota composition along larvae development.

This critique influences how I view the whole paper, it seems like you are reading into some data points and ignoring others. A rigorous data analysis would alleviate this concern.

We thank reviewer 1 criticism and hope that our comments helped to clarify the points raised.

I have other issues such as the use of different scales on the y-axis in your figures- it is really confusing. Consider using a standardized scale for gene expression, but this is a minor edit.

We created the gene expression graphs aiming to optimize the use of the graph area. When showing fold change in gene expression values that vary from 1 (control group) to 10e2 or 10e3, it is quite challenging to apply a single scale to all graphs. We thank reviewer 1 for highlighting this aspect, and we invested a significant time on this matter, but we could not find a common graph scale to fit to all graphs in a figure.

Reviewer 2 Report

  1. All scientific names described on pages 5, 8, and 10 must be changed to italics.
  2. The manuscript needs an English revision, and minor spelling changes should consider (example change “ddition” to “addition” on page 10. Etc. 3. Correct the following on page 6:
    - Def2 showed no significant difference when compared to non-fed-females (Figure 2-J).
    - Bacteria loads increased in females after feeding on sucrose at 24 and 48h PF (Figure 2-N).

Questions related to the strategies used:

  1. How do authors ensure that the selected universal primers (16S ribosomal) effectively allow amplifying all bacterial DNA (of all species)?
  2. Is there any additional control gene that allows the authors to support that the universal primers used in this manuscript allow all bacterial DNA amplification?
  3. Are there specific genes that allow differentiating the stages of Larvae and PP? Or the authors only make the differentiation of the stages macroscopically (using the microscope).

I had the last question because the authors indicate that the PP stage can empty its contents from the midgut, affecting the measurement of the bacterial content evaluated in this manuscript.

4. Is there any differences in motif or domains present in all defensin genes (by in silico analysis)?

5. How the authors identify all defensin genes (3 and 4) by DNA sequencing? We do not have that information in the manuscript.

6. The manuscript does not indicate the results obtained with positive and negative controls for the RNAi (gene silencing for the genes Att and Def2).

I consider that after the modifications and clarifications requested, the manuscript should be accepted.

Author Response

Responses to Reviewer #2

  1. All scientific names described on pages 5, 8, and 10 must be changed to italics.

The scientific names were corrected accordingly.

  1. The manuscript needs an English revision, and minor spelling changes should consider (example change “ddition” to “addition” on page 10. Etc.

We corrected the spelling and we performed language revision

  1. Correct the following on page 6:

    - Def2 showed no significant difference when compared to non-fed-females (Figure 2-J) (3.3).

Results on Def1 (Figure 2-I) and Def2 (Figure 2-J) were corrected for clarity.

    - Bacteria loads increased in females after feeding on sucrose at 24 and 48h PF (Figure 2-N).

Correction made following reviewer’s comment (3.3).

Questions related to the strategies used:

  1. How do authors ensure that the selected universal primers (16S ribosomal) effectively allow amplifying all bacterial DNA (of all species)?

16s primers are normally used in similar studies and amplify the majority of bacteria that compose the insect microbiota. We are aware that we may be missing some bacteria species since no single PCR primer pair has been described as the gold standard for 16S rRNA gene-based bacterial detection. Therefore, we cannot ensure that the 16S primers used in the present work effectively amplify all bacterial DNA. These primers amplify Eubacteria, which encompasses all the bacteria with the exception of Arqueobacteria.

In other to avoid any misunderstanding we removed the term “universal” from our results description and legends on pages 6 to 10.

  1. Is there any additional control gene that allows the authors to support that the universal primers used in this manuscript allow all bacterial DNA amplification?

We did not attempt to amplify all bacterial DNA. As mentioned above, we are aware that 16S primers cover Eubacteria.

  1. Are there specific genes that allow differentiating the stages of Larvae and PP? Or the authors only make the differentiation of the stages macroscopically (using the microscope).

I had the last question because the authors indicate that the PP stage can empty its contents from the midgut, affecting the measurement of the bacterial content evaluated in this manuscript.

Different developmental stages of sand flies are visually identified using microscope [1]. Under colony rearing conditions, sand fly larvae are constantly feeding on a dark substrate which makes their intestines visible through their cuticle. Under optimal rearing conditions only pre-pupal stage stops eating, intestines are emptied, and the whole larvae body is seen as light colored. No molecular markers are necessary for this kind of identification.

  1. Is there any differences in motif or domains present in all defensin genes (by in silico analysis)?

Among the four defensins identified in the present work, all four defensins have a motif belonging to the Defensin-2 superfamily. This information was added to the manuscript, on 3.1.

  1. How the authors identify all defensin genes (3 and 4) by DNA sequencing? We do not have that information in the manuscript.

On materials and methods 2.1, we described how Def3 and Def4 were identified: “Defensin 3 (Def3) (MW269862) and defensin 4 (Def4) (MW269863) were identified by using hidden Markov models (HMMER) of insect defensins as a query for a blastp analysis on an unassembled genomic sequences data base of L. longipalpis (GenBank PRJNA20279).”

  1. The manuscript does not indicate the results obtained with positive and negative controls for the RNAi (gene silencing for the genes Att and Def2).

The negative control for gene silencing is the group injected with dsLacZ which was represented as fold change 1 (horizontal line). The test dsRNAs injection groups are compared to this control. The actual silencing of the genes (which the reviewer is nominating as positive control) is verified by the expression of the corresponding gene (Att or Def2) after the dsRNA injection (shown in Figure S5, where a decreased expression is seen after silencing). The microinjection technique for dsRNA delivery was shown to be quite efficient in sand flies in ours and other labs [2–5].

I consider that after the modifications and clarifications requested, the manuscript should be accepted.

We thank Reviewer#2 for raising important questions and list the referenced studies below:

  1. Lawyer P, Killick-Kendrick M, Rowland T, Rowton E, Volf P. Laboratory colonization and mass rearing of phlebotomine sand flies (Diptera, Psychodidae). Parasite [Internet]. 2017/11/16. 2017;24:42.
  2. Sant’Anna MR, Alexander B, Bates PA, Dillon RJ. Gene silencing in phlebotomine sand flies: Xanthine dehydrogenase knock down by dsRNA microinjections. Insect Biochem Mol Biol [Internet]. 2008/05/31. 2008;38:652–60.
  3. Sant’anna MR, Diaz-Albiter H, Mubaraki M, Dillon RJ, Bates PA. Inhibition of trypsin expression in Lutzomyia longipalpis using RNAi enhances the survival of Leishmania. Parasit Vectors [Internet]. 2009/12/17. 2009;2:62.
  4. Telleria EL, Sant’Anna MRV, Ortigão-Farias JR, Pitaluga AN, Dillon VM, Bates PA, et al. Caspar-like gene depletion reduces leishmania infection in sand fly host Lutzomyia longipalpis. J Biol Chem. 2012;287:12985–93.
  5. Di-Blasi T, Telleria EL, Marques C, De Macedo Couto R, Da Silva-Neves M, Jancarova M, et al. Lutzomyia longipalpistgf-β has a role in leishmania infantum chagasisurvival in the vector. Front Cell Infect Microbiol. 2019;9:71.

Reviewer 3 Report

Despite the topic is not so popular, this is a well conducted experimental design with a large amount of data demonstrating the differences between the two differently fed groups. Discussion paragraph and introduction rely on a well described background. Supplementary material is suitable. Just few corrections before approval:

paragraph 2.1: change to "Similarities of L. longipalpis AMPs amino acid sequences with other insects were assessed"

paragraph 3.1: please italicize the species names and change to "that contain the six conserved cysteine residues (Figure S3) characterizing these AMPs"

paragraphs 3.4 and 3.5 and 3.6: please italicize the species names

paragraph 4: correct "in ddition" and change to "This timepoint corresponds to"

Author Response

Responses to Reviewer #3

Despite the topic is not so popular, this is a well conducted experimental design with a large amount of data demonstrating the differences between the two differently fed groups. Discussion paragraph and introduction rely on a well described background. Supplementary material is suitable. Just few corrections before approval:

Indeed, there is a small number of research groups focusing on sand fly immunity. Nevertheless, it reveals the need of adding contributions on this research topic.

We thank Reviewer#3 for the positive evaluation on our experimental design and manuscript sections.

paragraph 2.1: change to "Similarities of L. longipalpis AMPs amino acid sequences with other insects were assessed"

We made the correction following reviewer’s comment.

paragraph 3.1: please italicize the species names and change to "that contain the six conserved cysteine residues (Figure S3) characterizing these AMPs"

We made the correction following reviewer’s comment.

paragraphs 3.4 and 3.5 and 3.6: please italicize the species names

We made the correction following reviewer’s comment.

paragraph 4: correct "in ddition" and change to "This timepoint corresponds to"

We made the correction following reviewer’s comment.

Round 2

Reviewer 1 Report

My concerns have been addressed and this manuscript is now suitable for publication